# Brief communication: Measuring and modelling the ice thickness of the Grigoriev ice cap (Kyrgyzstan) and comparison with global datasets.

Lander VAN TRICHT[1], Chloë Marie PAICE[1], Oleg RYBAK[1,2,3], Philippe HUYBRECHTS[1]

[1]Earth System Science & Departement Geografie, Vrije Universiteit Brussel, Pleinlaan 2, B–1050 Brussels, Belgium
[2]Water Problems Institute, Russian Academy of Sciences, ul. Gubkina 3, Moscow, 119333 Russia
[3]FRC SSC RAS, ul. Ya. Fabritsiusa 2/28, Sochi, 354002 Russia

*Corresponding author: Lander Van Tricht (lander.van.tricht@vub.be)*

**Abstract.** An accurate ice thickness distribution is crucial for correct projections of the future state of an ice mass. However, measuring the ice thickness with an in-situ system is time-consuming and not scalable. Therefore, models have been developed to estimate the ice thickness without direct measurements. In this study, we reconstruct the ice thickness of the Grigoriev ice cap, Kyrgyzstan, from in-situ observations and the yield stress method. We compare the results with data from 6 global ice thickness datasets composed without the use of our local measurements. The results highlight the limitations of these generic datasets primarily stemming from the subdivision of ice caps into distinct glaciers, the adoption of a (calibrated) creep parameter value, assumptions regarding ice mass flux, and errors regarding surface velocity observations. These shortcomings especially emphasise the importance of integrating local observations to calibrate models to achieve precise representations of ice thickness, particularly when dealing with smaller or slow-flowing cold ice caps, such as the Grigoriev ice cap.

## 1. Introduction

The ice thickness distribution is an essential element in glaciological modelling studies as it represents the initial conditions of a glacier or ice cap in a model (Farinotti et al., 2017). To make projections about the future evolution of geometry and runoff, a correct representation of ice thickness and volume is thus essential. Because ice thickness field campaigns are often dangerous and time-consuming, detailed thickness data or distributions based on in situ measurements (e.g. radio-echo soundings), have only been obtained on just over a thousand glaciers and ice caps of the >200,000 remaining worldwide (Clarke et al., 2009; Welty et al., 2020). The aim of this brief communication is to present our measurements and reconstructed ice thickness distribution of the Grigoriev ice cap. During our multi-day field campaign in 2021, we measured the ice thickness at > 500 points using Radio Echo Sounding (RES). These radar measurements were converted into ice thickness and subsequently interpolated to the entire ice cap using an approach based on the yield stress. In addition, we compare the obtained ice thickness field with the reconstructed thickness from six global datasets composed without the use of our in-situ measurements (Farinotti et al., 2019; Millan et al., 2022).

## 2. Grigoriev ice cap

The Grigoriev ice cap (Figure 1) is located in the Inner Tien Shan (Kyrgyzstan, Central Asia) on the southern slopes of the Terskey Ala-Too mountain range, about 30 km northeast of the Kumtor Gold Mine and the Ak-Shyirak massif. The nearly circular ice cap, which is also called "a flat top glacier", has an altitude between 4200 and 4600 m a.s.l and covers an area of 7.5 km$^2$ (in August 2021). It is subject to a continental climate with a limited amount of precipitation, as the area is surrounded by high mountain ranges which protect the glaciers from incoming moisture. At the Kumtor-Tien Shan weather station (3659 m a.s.l.), the total annual precipitation is only 350 mm (Van Tricht et al., 2021). Most of the precipitation falls in spring and summer (75%), primarily as a result of local convection. In winter, the Siberian High with accompanying dry conditions rules over the region. The Grigoriev ice cap is thus an example of a spring/summer accumulation type of ice mass. In the past, several

glaciological measurements were performed on the ice cap, such as ice temperature measurements (Dikikh, 1965; Thompson et al., 1993; Arkhipov et al., 2004; Takeuchi et al., 2014) and surface mass balance measurements (Mikhalenko, 1989; Dyurgerov, 2002; Arkhipov et al., 2004; Fuijita et al., 2011). According to the modelling study by Van Tricht and Huybrechts (2022), the ice cap has a cold thermal regime.

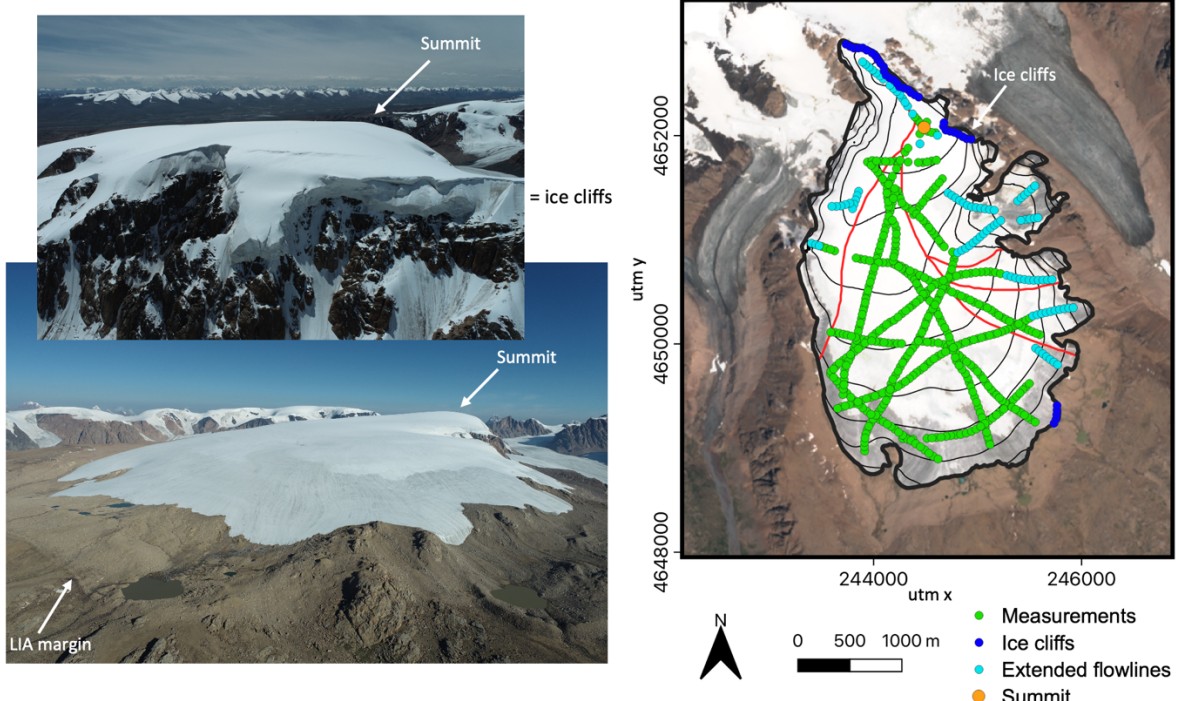

**Figure 1**: (left) View over the Grigoriev ice cap and the ice cliffs in August 2021. Both images are made with a DJI Phantom 4 RTK. (right) Grigoriev ice cap in August 2021. The background is from Sentinel-2 in July 2021. The elevation contours are drawn for every 50 m, starting from 4200 m a.s.l. The black outline of the ice cap is from August 2021. The coordinate system corresponds to the EPSG:32644 WGS 84 / UTM zone 44N. The red lines are the boundaries between the different parts of the ice cap in the Randolph Glacier Inventory version 6.

## 3. Measurements and modelling

### 3.1. Ice thickness measurements and drone data

The use of a radar or RES system to derive the ice thickness is based on the difference in permeability between ice and the underlying bedrock. As an electromagnetic wave travels more easily through ice than through bedrock, it will be reflected by the bedrock. Based on the difference in travel time between this reflected wave and the direct wave through the air, the ice thickness can be inferred (Figure 2) (Eq. 1):

$$H = \frac{1}{2} * \left[ v_{ice}^2 \left( \Delta t + \frac{d}{v_{air}} \right)^2 - d^2 \right]^{\frac{1}{2}} \tag{1}$$

with $v_{ice}$ the velocity of the wave through ice, assumed to be $1.68 \times 10^8$ m s$^{-1}$, and $v_{air}$ the velocity of the wave through air, equal to $3.00 \times 10^8$ m s$^{-1}$. H is the ice thickness, $\Delta t$ the time difference between the reception of both waves, and d is the physical distance between the transmitter of the wave and the receiver (typically 30-40 m).

In August 2021, we performed a multi-day field campaign on the Grigoriev ice cap to measure the ice thickness at more than 500 locations with a handheld ground penetrating radar (Narod and Clarke, 1994) (Figure 1). The identification of the bed reflection consisted of a manual process in which the position of the reflected wave was precisely marked on the radargram (Figure 2). With the time difference between the reception of this

reflected wave and the direct wave, the signal was then translated into the local ice thickness using equation 1. Furthermore, a post-processing migration technique was applied to remove improbable measurements (Binder et al., 2009; Andreassen et al., 2015). However, this migration procedure did not lead to any modifications in the derived ice thickness values. Following the setup of previous field campaigns (Van Tricht et al., 2021a), a radio signal with a frequency of 5 MHz was chosen for all measurements.

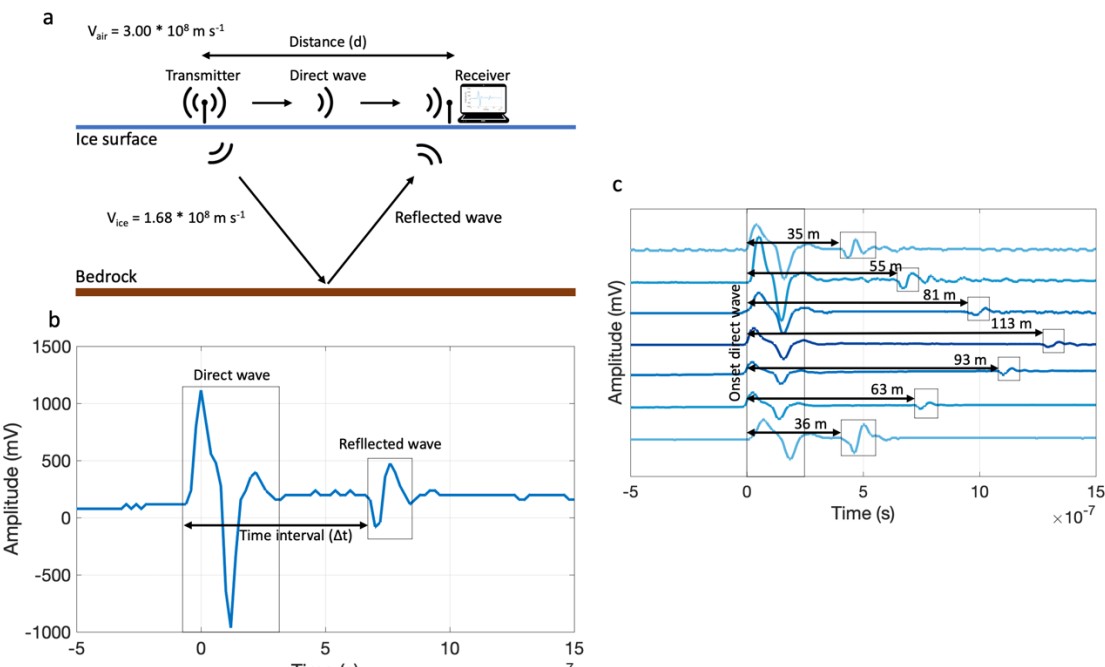

**Figure 2**: (a) Schematic setup of the measurements. (b) example of a reflected signal used to infer the ice thickness. (c) Seven different radar profiles with their associated ice thickness.

Using the approach of Van Tricht et al. (2021a), the uncertainty of the ice thickness measurements is estimated to be 8 m $\pm$ 0.05*H. GPS measurements of the locations of the transmitter and the receiver were made with a TRIMBLE GeoX7 and differentially corrected afterwards using the nearby base station of Kumtor, resulting in a typical horizontal precision of 0.1-0.2 m and a vertical precision of 0.2-0.3 m. In addition to the radar measurements, a DJI Phantom 4 RTK drone was used to capture > 1000 images to reconstruct the surface

elevation of the ice cap using the photogrammetry workflow in Pix4D (Van Tricht et al., 2021c). During the drone surveys, a total of 42 orange plastic squares of 30x30 cm were strategically distributed as ground control points (GCPs) across the glacier's surface, and at some exposed bedrock sites near the ice margin. Accurate positions of these GCPs were established using the GPS device and subsequently utilised for georeferencing and validation purposes. The validation yielded a root mean square error (RMSE) of 0.06 m horizontally and 0.09 m vertically

indicating a very high accuracy of the 2021 DEM.

### 3.2. Yield stress method

     Due to time and safety constraints, not all parts of the ice cap could be covered with measurements. Therefore,

to complement the interpolation procedure (section 3.3), the yield stress method is employed to partly fill in the gaps (Figure 1). This method assumes perfect plasticity (Linsbauer et al., 2012; Li et al., 2012; Zekollari et al., 2013). The assumption is that the yield stress ($\tau_y$) (~ basal shear stress) can be determined for measured points based on the local ice thickness and the local surface slope (Eq. 2) and that the mean yield stress can be assigned to unmeasured locations to infer the ice thickness along flowlines.

$$\tau_y = \rho \, g \, H \sin \alpha \tag{2}$$

$\alpha$ is the local surface slope averaged over a 250x250 m square and $\rho$ is the average ice density (900 kg/m³). As the Grigoriev ice cap is not surrounded by valley walls, a shape factor to account for lateral drag is not included here (Li et al., 2012; Pieczonka et al., 2018). However, since a large part of the ice cap was accessible for measurements, we opted not to assign the mean yield stress, but to interpolate the yield stress over the ice cap and assign the obtained value ($\tau_y^*$) to several individual points at the position of unmeasured flowlines (Figure 1). Subsequently, the local ice thickness for these additional points (in total 94 points, mainly at the eastern outlet glaciers) was inferred (Eq. 3):

$$H = \frac{\tau_y^*}{\rho \, g \, \sin \alpha} \tag{3}$$

Previous studies (Li et al., 2012; Farinotti et al., 2017) showed that Eq. 3 tends to overestimate the ice thickness in very flat regions (small slope). Therefore, we implemented a minimum slope of 5% and only determined the ice thickness for points with larger slopes (Pieczonka et al., 2018). We also derived the mean yield stress based on all measurements ($\tau_y$ from Eq. 2), which appeared to be 73.3 kPa. This matches quite closely with the basal shear stress of 78.88 kPa determined from the empirical relationship between average basal shear stress and the elevation range of the glacier, described in Haeberli and Hoelzle (1995).

### 3.3. ANUDEM interpolation

In addition to all measurements and reconstructed ice thickness points along flowlines, as a boundary condition, the ice thickness along the margin of the ice cap was set to 5 m, which is a realistic assumption for grid points situated at 12.5 m from the margin (~ half horizontal resolution) (Zekollari et al., 2013) and 0 m outside the glacier area. However, the Grigoriev ice cap is also characterised by dry calving cliffs at the northern margin (Figure 1). Therefore, the ice thickness along this part was manually adjusted based on the elevation difference between the ice margin and the bedrock next to it. Finally, to achieve a full ice thickness distribution of the ice cap, all ice thickness data were interpolated to the entire ice cap using the ANUDEM algorithm, developed by Hutchinson (1989), which has been widely employed for ice thickness interpolation in previous studies (e.g., Fischer, 2009; Linsbauer et al., 2012; Van Tricht et al., 2021a). The algorithm was applied using the Topo-To-Raster tool. The resolution of the final ice thickness distribution was set to 25 m.

## 4. Results and discussion

### 4.1. Measured ice thickness and estimated volume

During the field campaign, the ice thickness was successfully determined for 481 locations. For ca. 30 locations, no clear ice thickness could be determined because of distortions in the waveform. The mean measured ice thickness appeared to be 73.05 ± 11.65 m, while the maximal measured ice thickness was 114.85 ± 13.74 m. Takeuchi et al. (2014) found an ice thickness of 86.87 m for the ice core that was taken in 2007 near the summit of the Grigoriev ice cap. For the location of the ice core, we found a thickness of 78.30 ± 11.91 m (difference of ca. -8 m), which is within the error bounds. However, a potential cause for the difference could be thinning of the ice at the summit between 2007 and 2021. A comparison between the elevation of the drilling site in 2007 derived from GPS measurements and the corresponding elevation of this site in 2021 revealed a slight lowering of the surface (-1.32 m) over the past 14 years. Another reason for the mismatch might be explained by the assumed constant velocity of the radar wave used to infer the ice thickness. The velocity was assumed to be constant at $1.68 \times 10^8$ m s⁻¹, which is the travel velocity for pure ice. However, layers of snow and firn were detected in the upper 22 m of the ice core (Takeuchi et al., 2014), which can lead to an underestimation of the ice thickness when using this constant travel velocity. Using the density profile of the ice core acquired in 2007, we calculated an average radar wave velocity of $1.75 \times 10^8$ m s⁻¹. Employing this velocity in Eq. 1 would raise the

160 measured ice thickness by 3.31 m at the ice core site. By accounting for both corrections (thinning of the summit and higher radar wave velocity), the resulting measured ice thickness becomes to be 82.93 ± 12.14 m (in 2007). This value closely aligns with the ice thickness obtained directly from the ice core. After interpolation of all the ice thickness data, a total ice volume of 0.392 (0.312 - 0.473) km³ was derived (Figure 3a).

## 4.2. Comparison with global ice thickness and volume estimates

We compare our results with six existing ice thickness distributions and volume estimates composed without in-situ data (Figure 3b-g). The five different ice thickness distributions presented in Farinotti et al. (2019) as well as the Millan dataset (Millan et al., 2022) were constructed using the SRTM DEM to compute the surface slope, 170 principles of ice flow dynamics and the Randolph Glacier Inventory (RGI) v6.0 outline (RGI, 2017). In this inventory, the Grigoriev ice cap is subdivided into five separated branches (Figure 1) based on an algorithm for detection of ice divides (Pfeffer et al., 2014).

To ensure a meaningful comparison, we reconstruct the ice thickness distribution for the consensus estimate 175 and models 1-4 by accounting for surface elevation changes between 2002 (retrieved from SRTM data) and 2021 (derived from the UAV DEM data). Regarding the Millan dataset, we only need to account for the elevation changes between 2018 and 2021, as the ice thickness in this dataset was inferred from the 2017/2018 surface velocities, obtained from satellite images using the Shallow Ice Approximation (SIA) and the SRTM DEM surface slope. We therefore assume it to be representative for 2017/2018. To maintain consistency and avoid potential 180 errors associated with the geometry of different years, we limit our analysis to the glaciated area in 2021 for all comparisons.

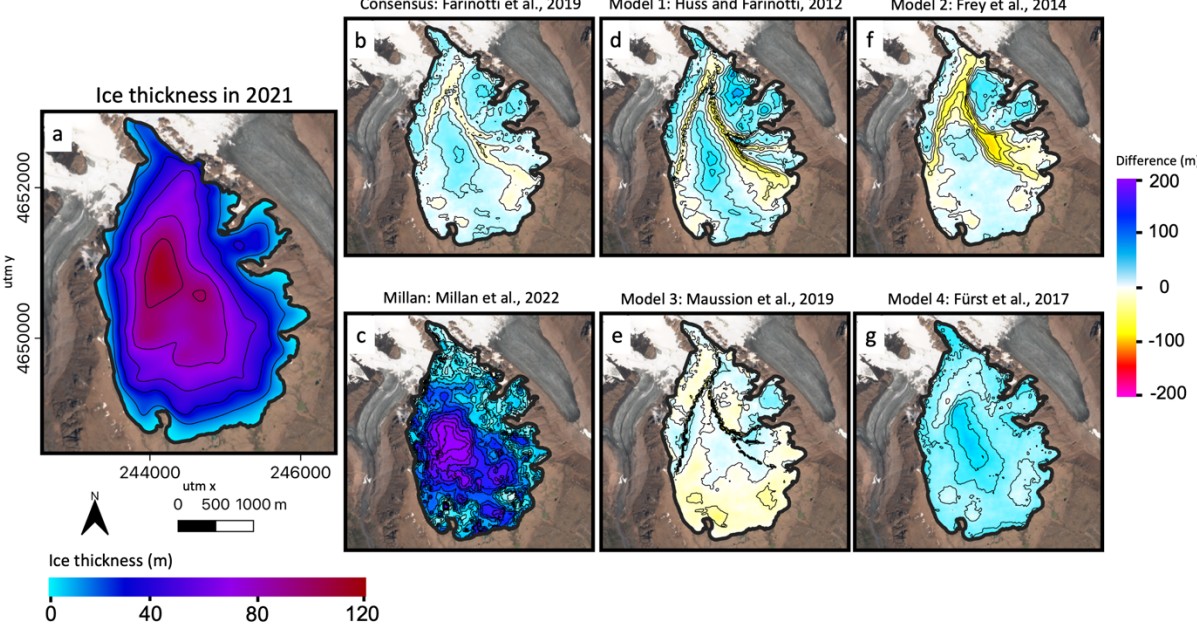

**Figure 3**: (a) Ice thickness of the Grigoriev ice cap in August 2021. The coordinate system corresponds to the EPSG:32644 185 WGS 84 / UTM zone 44N. (b-g) Difference between the created ice thickness distribution and the consensus estimate (b), the different models used to compile the consensus estimate (d,e,f,g) and the Millan dataset (c). The different ice thickness datasets are corrected to represent the state in 2021 for a proper comparison with our own reconstruction (panel a). The background of the seven panels is from Sentinel-2 in July 2021. Contours are added for every 20 m.

Model 1 (Huss and Farinotti, 2012), model 3 (Maussion et al., 2019), and model 4 (Fürst et al., 2017) operate on the fundamental principle of mass conservation to assess the glacier's mass turnover by estimating the

distribution of mass balance and elevation changes. These models calculate the mass flux and subsequently convert it into ice thickness using a prescribed constitutive relation (in the case of model 1 and model 3) or by employing an ice flow model (in model 4). The primary distinction between model 1 and model 3 lies in their approaches to compute the mass balance. Model 1 prescribes the mass balance as a linear function of elevation and continentality, while model 3 employs a temperature-index model driven by gridded climate data to simulate the mass balance. Moreover, while model 3 generates multiple flowlines to represent the glacier's flow, model 1 simplifies the representation by compressing the glacier into two-dimensional elevation bins. Subsequently, both models extrapolate data from a 2-D representation of the glacier (Model 1) or the thickness distribution along the flowline (Model 3) to a comprehensive grid encompassing the entire glacier. Model 4 of Fürst et al. (2017) consists of a minimisation approach based on mass conservation to derive glacier ice thickness. This model uses distributed fields of surface mass balance, obtained from the GloGEM model by Huss and Hock (2015), and the rate of ice thickness change, obtained from a parametrisation based on glacier size by Huss et al. (2010). The mass conservation equation is solved using Elmer/Ice software, and the resulting flux solution is then translated into a glacier-wide thickness field, using the SIA.

Model 2 (GlabTop2) of Frey et al. (2014) adopts a shear/yield-stress-based approach for ice thickness modelling. This method relies on an empirical relationship between average basal shear stress and the elevation range of the glacier, as found by Haeberli and Hoelzle (1995), in order to compute the ice thickness at specific locations using the SIA. Subsequently the ice thickness is interpolated across the entire glacier.

In general, model 1 exhibits thicker ice compared to our reconstruction (Figure 3d, Table 1), except at the boundaries of the RGI individual units where the ice thickness is constrained to 0 meters. The RGI segmentation of different parts of the ice cap introduces evident boundary effects, which is also the case for model 2 and model 3. The slight overestimation of the ice thickness in model 1 can likely be attributed to two factors. Firstly, the ice flux may be overestimated due to a too large surface mass balance (SMB) gradient. A study conducted by Van Tricht and Huybrechts (2022) demonstrated that the Grigoriev ice cap area is associated with a local very low precipitation gradient. This leads to a smaller mass balance gradient compared to other glaciers in the vicinity of the ice cap. In addition, Huss and Farinotti (2012) prescribe the mass balance gradient as a function of continentality, which is regionally uniform. This also suggests that the mass balance gradient employed in model 1 might be too large, thus resulting in an increased ice mass flux and generally thicker ice. Secondly, the creep parameter used to determine the ice thickness might be too low. Huss and Farinotti (2012) calculate the temperature-dependent creep parameter by assuming a constant offset of 7°C between the average ice temperature of the glacier and the temperature at the equilibrium line altitude (ELA). Following this approach, a mean annual air temperature of -10°C at the ELA is obtained, corresponding to an englacial temperature of -17°C. This yields a very low creep parameter of $2.5 \times 10^{-17}$ $Pa^{-3}$ $yr^{-1}$ in their approach. However, Van Tricht and Huybrechts (2022) found that the mean ice temperature of the Grigoriev ice cap is -4.2°C, which would correspond to a higher creep parameter of $4.6 \times 10^{-17}$ $Pa^{-3}$ $yr^{-1}$. Using the latter value in the formulas of Huss and Farinotti (2012) would result in a lower reconstructed ice thickness.

**Table 1.** Volume and maximum ice thickness of the Grigoriev ice cap in 2021 according to the different ice thickness distributions. The root mean squared error (RMSE) and mean error (ME) are calculated by comparing the modelled ice thickness with the in-situ measurements.

|  | Measurements | Consensus | Millan | Model 1 | Model 2 | Model 3 | Model 4 |
|---|---|---|---|---|---|---|---|
| Vol (km³) | 0.392 | 0.485 | 1.155 | 0.494 | 0.403 | 0.377 | 0.640 |
| $H_{max}$ (m) | 114 | 147 | 359 | 163 | 137 | 131 | 187 |
| RMSE (m) |  | 19.70 | 141.35 | 26.25 | 24.42 | 16.13 | 42.08 |
| ME (m) |  | 12.00 | 130.21 | 11.31 | -1.75 | -3.78 | 36.96 |

Our study found that the average yield stress derived from our measurements, 73.3 kPa, closely matches the yield stress value of 78.9 kPa obtained from the empirical formula by Haeberli and Hoelzle (1995), which was

used in model 2. Generally, model 2 exhibits a slight overestimation in ice thickness (Figure 3f and Table 1), which can likely be attributed to this slightly higher yield stress used in the model. However, like model 1 and model 3, discrepancies mainly arise near the boundaries of the RGI glaciers, where the interpolation scheme of model 2 assigns a minimum ice thickness to ensure realistic glacier cross-sections.

Model 3 generally exhibits thinner ice, particularly noticeable at the ice cap's front (Figure 3e), resulting in a slightly reduced total volume for the year 2021 in comparison to our reconstruction (Table 1). Moreover, the boundary effects observed at the margins of the RGI glaciers are less pronounced in this model. The reduced ice thickness at the ice cap's front, as compared to our observations and reconstruction, could potentially be attributed to a high creep parameter used in the model. Maussion et al. (2019) used a default value of $7.6 \times 10^{-17}$ $Pa^{-3}$ $yr^{-1}$ for this parameter, which is a typical value for temperate glaciers. However, the Grigoriev ice cap is a cold ice cap (Van Tricht and Huybrechts, 2022), which is associated with a lower creep parameter. The same phenomenon was observed for the Urumqi glacier, a cold glacier located in the eastern Tien Shan (Farinotti et al., 2017), for which the modelled ice thickness was found to be too thin compared to actual observations. Nevertheless, among all model results (Table 1), model 3 matches most closely with our observations.

Notably, Model 4 does not exhibit the boundary effects of the RGI parts because it does not enforce the ice thickness to reach zero at the margin. In contrast, internal boundaries are dissolved, and the ice thickness solution is computed for glacier compounds. However, model 4 significantly overestimates the ice thickness (Figure 3g), leading also to a high RMSE and ME with respect to our measurements (Table 1). As for model 1, this overestimation can likely be related to a too large ice flux or a too low creep factor. Model 4 typically employs all available thickness measurements per RGI region to determine a region-uniform viscosity value. During the analysis, the lack of direct measurements in the vicinity of the Grigoriev ice cap in the GlaThiDa database resulted in using ice viscosity values based on measurements from glaciers located further away, possibly leading to an underestimation of the viscosity value for the Grigoriev ice cap.

The consensus estimate represents a composite solution achieved through a weighted combination of the outcomes obtained from models 1-4. It is clearly an intermediate solution, positioned between the more extreme results provided by the individual models (Figure 3b). While the consensus estimate captures the overall pattern of ice thickness, it tends to generally overestimate the ice thickness, primarily due to the contributions from model 1 and model 4. Besides, the boundary effects of the RGI are still conspicuously present in this combined solution.

Lastly, as can be seen, the ice thickness of the Millan dataset is significantly larger than our reconstructed ice thickness field (Figure 3c). For the larger part of the ice cap, the Millan et al. (2022) estimate is between two to four times larger than the measured ice thickness. For instance, the maximum ice thickness of the Millan dataset is 350 m, while we measured a maximum of 114 m $\pm$ 13.74 m. Regarding volume, the Millan dataset presents a value of 1.155 km³ in 2021, which is 2.9x larger than our reconstructed volume. The significantly thicker ice in the Millan dataset is mainly related to an overestimation of the surface velocity. By comparing observed (from stakes) and modelled velocities with the velocities of Millan et al. (2022), we find a very large discrepancy. For the thickest part of the ice cap, the Millan velocity map indicates velocities up to 80 m $yr^{-1}$ while the stake and model derived velocities are of the order of 3-5 m $yr^{-1}$ (Van Tricht and Huybrechts, 2022). We hypothesise that the velocities of this slowly moving ice cap have been substantially overestimated due to the presence of snow at the surface during most of the year, leading to low contrasts and an absence of features to trace over the year. Furthermore, Millan et al. (2022) used an average creep parameter of $7.2 \times 10^{-18}$ $Pa^{-3}$ $yr^{-1}$ for the region of the Grigoriev ice cap. This value is the lowest of all regions included in their study, and equal to the value of the region southeast of the ice cap as no ice thickness data were available in the GlaThiDa at the time of the analysis. Such a low creep parameter value also contributes to larger ice thickness.

## 5. Conclusions

In this study, we measured and modelled the ice thickness of the Grigoriev ice cap in the Inner Tien Shan, Kyrgyzstan, and we compared the obtained ice thickness distribution with the results from 6 global ice thickness datasets. The main take-away from the analysis is that the global datasets do not perform well enough yet for ice caps such as the Grigoriev ice cap. Discrepancies between our observations and the consensus estimate of Farinotti et al. (2019), as well as the individual models from which it was composed, are mainly caused by the division of the ice caps into multiple glaciers, the value of the creep parameter, ice flux assumptions, and the dominance of temperate valley glaciers in the calibration of the models. These weaknesses were already mentioned earlier (Farinotti et al., 2017). The newest dataset by Millan et al. (2022), which relies on surface velocity observations, effectively captures the pattern of ice thickness and exhibits no boundary effects at ice divides. Yet, it significantly overestimates the ice thickness, mainly due to the overestimation of the surface velocities. Consequently, our results underscore the continued necessity of local ice thickness measurements to achieve accurate representations of ice thickness and volume estimates, particularly for smaller or slow-flowing cold ice caps such as the Grigoriev ice cap. Moreover, for ice caps, improved ice thickness estimates near ice divides could be achieved by avoiding ice mass subdivision. Additionally, incorporating supplementary information, such as accurate surface ice flow velocities, surface mass balance gradients or a creep parameter adapted to the thermal regime of the considered ice mass, could enhance the reliability of ice thickness estimates, as many methods rely on ice flux estimations. In summary, it thus remains crucial to recognise that the adoption of global ice thickness datasets can have significant implications, especially at the local scale, for projecting future ice volume and the associated evolution of runoff.

## 6. Data availability

Research data and results are provided through an online public repository, accessible via https://zenodo.org/badge/latestdoi/614248752 (Van Tricht, 2023). Information and specific details about the model code will be specified on request by Lander Van Tricht. The ice thickness measurements will be provided to the GlaThiDa (https://www.gtn-g.ch/glathida/).

## 7. Author contribution

All authors contributed to the fieldwork. LVT conducted the research and wrote the manuscript with help from PH and CMP. OR organised the fieldwork. We also specifically want to thank Benjamin Vanbiervliet, who assisted during the field campaign and helped to analyse the preliminary data.

## 8. Competing interests

The authors declare that they have no conflict of interest.

## 9. Acknowledgements

We would like to thank everyone who contributed to the fieldwork.

## 10. Financial Support

Lander Van Tricht holds a PhD fellowship of the Research Foundation-Flanders (FWO-Vlaanderen) and is affiliated with the Vrije Universiteit Brussel (VUB).

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
