# Peer review of "Brief communication: Measuring and modelling the ice thickness of the Grigoriev ice cap (Kyrgyzstan) and comparison with global datasets."

_EGUsphere, 2023_

## Referee Comment (RC1)

Review of "Brief communication: Measuring and modelling the ice thickness of the Grigoriev ice cap (Kyrgyzstan) and comparison with global datasets" by Van Tricht et al.

This brief communication presents new ground penetrating radar (GPR) ice thickness measurements for the Grigoriev Ice Cap in Kyrgyzstan. The manuscript describes the field campaign, the GPR measurements, and the interpolation method used to obtain a complete ice thickness map. Finally, the authors compare their results with global ice thickness datasets and highlight the discrepancies.

The figures are nice and the paper clear and well organized but the content of the paper is weak at this stage, even for a brief communication. Major revisions are required before it can be considered for publication.

**General comments**

Just saying that field measurements are needed because global thickness products are not accurate in this particular case is not very interesting in itself. Global products do not attempt to be accurate everywhere, but rather give a volume estimate on a regional to global scale. The specific case of a polythermal, small ice cap is exactly where one would expect global scale estimates to be wrong.

In my opinion, what would make this communication publishable would be to highlight the reason why the different global estimates do not reproduce the observations. This would allow to identify which assumption done in those estimates can be improved and how. In the current version of the manuscript, this work is poorly done, as the authors have not really looked in detail at how these global estimates are made. This is shown by their assumption that these estimates are done for the year 2002 due to the SRTM DEM, which is wrong. This leads to a wrong correction of their thickness field and to irrelevant comparisons. For example, Milan et al. uses surface velocity from 2017/2018 combined to the shallow ice approximation to provide thickness estimate. The SRTM DEM from 2002 is only used to compute the surface slope. The method and assumptions of each estimate presented should be reviewed and analyzed in the light of what is known about the Grigoriev Ice Cap. This would allow to identify the origin of the errors in the reconstructed thickness (mass balance, ice viscosity, sliding, surface velocity .....).

**Specific comments**

You will find a list of correction and specific comments embedded in the annotated PDF in attachment.

---

## Referee Comment (RC2)

[referee-annotated manuscript omitted]

---

## Referee Comment (RC3)

**General remarks**

Lander Van Tricht and colleagues present the ice thickness estimations of the Grigoriev ice cap (Kyrgyzstan) collected in several field campaigns during August 2021 (summer) by using GPR technique. Then, the radar data was processed by applying the yield stress method and interpolated to produce an ice thickness layer. Finally, the authors evaluate if the global outputs resulting from 6 different experiments are able to capture the spatial patterns of the ice thickness at the local scale in the Grigoriev ice cap.

The manuscript is well structured, clear, and concise, making it easy to understand. I congratulate the authors because they have compiled a large amount of data with potential for scientific applications, however, they do not give enough detail on the statistical approach demonstrating the unreliability of the global datasets, and it seems they remain in a visual description of the discrepancies.

The global ice thickness products were conceived as an approximation of the total volume of ice available on the Earth's surface, with its associated uncertainty. It is therefore logical to expect that their site-specific net representation will vary from site to site, depending on morpho-topographic conditions. In addition, several of the world's ice masses are inaccessible for logistical or risk reasons, in which case in-situ observations are simply not feasible. This does not seem to be the case with Grigoriev. Therefore, numerical modelling products can provide valuable complementary data to field measurements.

There is a methodological gap in this study and the authors need to work on major corrections before this manuscript can be published in TC.

**Detailed remarks**

L11. I am not sure how the under-representation of ice thickness in the global dataset demonstrates the importance of in-situ measurements. Please provide more evidence of the specific factors that may render the thickness data obtained by global models deficient, e.g. the role of basal topography. For example, it would be interesting to suggest methodological considerations that would improve model outputs.

L38. Grigoriev Ice Cap has a gentle topography, which allows most of the ice cap area to be covered by radar, but not so for other glaciers. Since data are available, please shed some light on the role of mass balance, dynamics and morphology in explaining such discrepancies. Also give the area covered by the ice cap.

L45. I don't think 'range' is the right word, but if it is, then provide a range of mass balance or thermal profile max/min values.

L74. The error estimate is not clear. Is it 8m or 5%? In the location, in the profile or in the interpolated area? Provide a detailed description of how do you arrive at these values or a reference.

L77. Give a description of the photogrammetric process. Did you perform a geodetic adjustment? If so, give the mean error in the horizontal and vertical residuals. How many ground control points did you use? Are GCPs located in the off-glacier areas?

L78. 0.2 m is the nominal uncertainty of the GPS or has the adjustment error been reached, please clarify

L90. Please show some radar profiles.

L111. Instead to interpolate $\tau_y$ why not only compute a bedrock surface model taking advantage of the high resolution data you have.

L128 and L133. Text is repeated

L136. I do not understand why you stick to visual inspection when you can make a robust statistical comparison of the two spatial datasets. This involves checking the spatial distribution, patterns, and correlations between spatial locations of ice thickness once the data are standardized.

Section 4.2. In view of the comment made by referee 1 about the misunderstanding of the approach used by Milan et al. and Farinotti et al., this comparison should be reviewed and adjusted to obtain a reliable interpretation.

Conclusions. If data are available, please shed some light on the role of mass balance, dynamics, and morphology in explaining such discrepancies. It could be interesting to identify the reasons for such discrepancies between datasets and to suggest approaches to resolve such discrepancies. For example, I would like to see the authors propose some alternatives for adjusting global products based on local observations or evaluate the representativeness of the global products in terms of their applications for estimating the future evolution of ice masses or runoff in the context of a changing climate.

---

## Author Response (AR1)

Dear Lander and co-authors,

Thank you for your response to reviewer comments and for your extensive edits, which I am satisfied address the concerns of the reviewers. I am happy to accept this manuscript for publication in the Cryosphere, however would ask that you please first consider the minor suggestions listed below.

Kind regards and thanks again for your submission.

Caroline

1. Please check that the added radar profiles in Figure 2 are appropriate for colorblind readers.
2. In the title for section 3.3 I believe "ANUDEM" should be capitalized. It should also be introduced within the text of this section when discussing the Topo-Ro-Raster algorithm.

*Dear editor,*

*We greatly appreciate your positive recommendation for publication of our manuscript. We provided the latest version of the manuscript with the two minor revisions you suggested. We opted to use a blue colour scale.*

*Kind regards*

*Lander Van Tricht*